# A pilot cost-benefit analysis of a children's spectacle reimbursement scheme: Evidence for Including children's spectacles in Mongolia's Social Health Insurance

Ai Chee Yong[1], Chimgee Chuluunkhuu[2], Ving Fai Chan[1,3]*, Tai Stephan[4], Nathan Congdon[1,4,5], Ciaran O'Neill[1]

1 Centre for Public Health, School of Medicine, Dentistry and Biological Sciences, Queen's University Belfast, Northern Ireland, United Kingdom, 2 Orbis North Asia, North Asia, Mongolia, 3 College of Health Sciences, University KwaZulu Natal, Durban, South Africa, 4 Orbis International, New York City, New York, United States of America, 5 Zhongshan Ophthalmic Centre, Sun Yat-sen University, Guangzhou, China

* v.chan@qub.ac.uk

**Data Availability Statement:** All relevant data are within the paper and its Supporting Information files.

## Abstract

### Background and aim

Globally, 12.8 million children have vision impairment due to uncorrected refractive error (URE). In Mongolia, one in five children needs but do not have access to spectacles. This pilot cost-benefit analysis aims to estimate the net benefits of a children's spectacles reimbursement scheme in Mongolia.

### Methods

A willingness-to-pay (WTP) survey using the contingent valuation method was administered to rural and urban Mongolia respondents. The survey assessed WTP in additional annual taxes for any child with refractive error to be provided government-subsidised spectacles. Net benefits were then calculated based on mean WTP (i.e. benefit) and cost of spectacles.

### Results

The survey recruited 50 respondents (mean age 40.2 ± 9.86 years; 78.0% women; 100% response rate) from rural and urban Mongolia. Mean WTP was US$24.00 ± 5.15 (95% CI US$22.55 to 25.35). The average cost of a pair of spectacles in Mongolia is US$15.00. Subtracting the average cost of spectacles from mean WTP yielded a mean positive net benefit of US$9.00.

### Conclusion

A spectacle reimbursement scheme is potentially a cost-effective intervention to address childhood vision impairment due to URE in Mongolia. These preliminary findings support the proposal of the inclusion of children's spectacles into existing Social Health Insurance. A

**Funding:** The authors received no specific funding for this work.

**Competing interests:** Professor Congdon declares that he is the Director of Research for Orbis International, Dr Chuluunkhuu declares that she is the Country Director for Orbis Mongolia, and Tai Stephan declares that she is the Global Programme Manager for Orbis International. This does not alter our adherence to PLOS ONE policies on sharing data and materials.

much larger random sample could be employed in future research to increase the precision and generalisability of findings.

## Introduction

Uncorrected refractive error (URE) accounts for 61% of the total global burden of vision impairment (VI) [1], and affects 12.8 million children worldwide [2]. Despite the existence of evidence-based and cost-effective strategies, limited affordability and access to high-quality refractive services remain the major barriers to better spectacles coverage in low- and middle-income countries [3–5]. Delivery of spectacles is shown to improve cognitive development, educational achievement, work productivity, and psychosocial well-being [6–10]. The annual global economic loss caused by VI due to URE is estimated to be US$202 billion [11].

Mongolia, the least densely populated nation on earth (2 people/km$^2$), is a landlocked country in North-Central Asia [12]. The Mongolian government provides citizens universal access to health care services covered under Social Health Insurance (SHI) [13], and children receive free at point of use health and dental care [14,15]. However, SHI does not cover spectacles, despite their inclusion on the World Health Organisation (WHO) Priority Assistive Product List [16]. According to the Mongolian Resolution for the National Non-communicable Disease Programme, more than 90% of Mongolian children with VI due to URE do not have spectacles [17]. Given the urgency to address the burden of URE among children, there is a natural interest in a children's spectacle reimbursement scheme for Mongolia.

Ready-made spectacles are offered with lenses of the same spherical prescription in both eyes, while custom spectacles can be offered with combination of prescriptions to correct any magnitude of refractive errors (RE). Studies suggest that low cost, ready-made spectacles are effective at correcting RE, without compromising spectacle wear while reducing costs and solving logistical challenges of school-based refractive service programmes [18,19]. The number of children who can benefit from a pair of ready-made spectacles is high in China [20], India [18], and Cambodia [21], which ranges from 70–83%. Based on a global dataset obtained from a screening programme supported by an international eye non-governmental organisation (NGO), 51.4% of Mongolian children were deemed clinically suitable for ready-made spectacles (OneSight, 2021).

To inform policymakers of the potential benefits of such a scheme, a cost-benefit analysis (CBA) is preferable over other health economic analyses because CBA reports outcomes in monetary terms, which are easily presented to decision-makers [22]. Langabeer et al.'s CBA on telemedicine demonstrates potential annual savings of US$928,000 in Houston, United States, when compared to traditional emergency medical services [23]. A willingness-to-pay (WTP) survey is one way in which preferences can be elicited for use in CBA that can hypothetically estimate an intervention's benefits [24]. It assesses how much a target population is willing to pay for an intervention. WTP has been used to estimate the potential value of a proposed spectacle delivery scheme in rural Cambodia [25].

The Mongolian SHI is largely funded by the state central budget through general taxation. Should the proposed child's spectacle reimbursement scheme be adopted by the government, the SHI would cover the reimbursement cost of eye examination and children's spectacles [26]. However, the actual framework for the reimbursement scheme is yet to be structured. The exploration of such a framework will be in our future scale-up study.

Despite a growing number of economic evaluation studies on URE programmes, few studies are on children. This pilot CBA is designed to assist eye health NGO Orbis International to

provide initial findings to the Mongolian policymakers of the potential benefits of a proposed child's spectacle reimbursement scheme.

## Materials and methods

This pilot study was approved by the Faculty of Medicine, Health and Life Sciences Research Ethics Committee, Queen's University Belfast (reference number MLHS 20_73). The study protocol was reviewed by the local gatekeeper, Mongolian Ophthalmologist's Society, and assured its adherence to the Mongolia's ethics regulations (reference number MOS_04). Verbal consent was obtained from each respondent upon agreeing to participate in the survey.

### Design and setting

A WTP survey of rural and urban Mongolia was used to estimate the benefits of providing spectacles to any children with RE. Those estimated benefits were used in the CBA.

### Sampling

This study used a trained local Mongolian-speaking enumerator to recruit 50 taxpayers, who were parents of children participating in a school-based vision screening programme conducted by Orbis International and OneSight (both are eye health NGOs). According to the central limit theorem for sample size of more than 30 [27], the sampling distribution was assumed to be normal. Upon discussions with the local researchers, we increased the sample size to 50 parents as a contingency to a high non-response rate. Parents from the Orbis contact list were randomly selected where samples were clustered into rural and urban schools. Criterion sampling was employed to recruit (i) 12 parents of children who do not need spectacles living in rural settings and 13 from urban settings, (ii) 12 parents of children who were provided with spectacles living in rural settings and 13 from urban settings.

### Willingness-to-pay survey

A triple-bounded-dichotomous-choice experiment (TBDC) was used to facilitate value elicitation by leading respondents logically through consideration of their WTP [28]. The market cost of a pair of spectacles in Mongolia was used to determine realistic starting bids. The average cost of a pair of ready-made spectacles of US$5.00 (MNT15,000) and the cost of custom spectacles of US$25.00 (MNT75,000) were used to inform bids [29]. After discussions with local eye care personnel as to their impressions of what might be reasonable, three starting bids were established: a low bid—US$12.50, medium bid—US$17.50, and high bid—US$22.50. The subsequent bids were dependent on the acceptance (Yes) or rejection (No) of the former bid. In the case of respondents offering no maximum limit, the maximum WTP amount of US$30.00 was taken. To reduce anchoring bias, the starting bid used to initiate the survey was randomly selected [30] (Fig 1).

### Costs of spectacles

Three costs for spectacles were used to calculate the scheme's net benefits: US$5.00 for ready-made spectacles, US$25.00 for custom spectacles, and US$15.00 for spectacles with an equal probability of being either. Based on a dataset obtained from an outreach vision screening programme initiated by an international eye NGO, one in two Mongolian children who had refractive error can be corrected effectively from a pair of inexpensive, ready-made spectacles. Therefore, the costs of ready-made spectacles and custom spectacles were used to construct

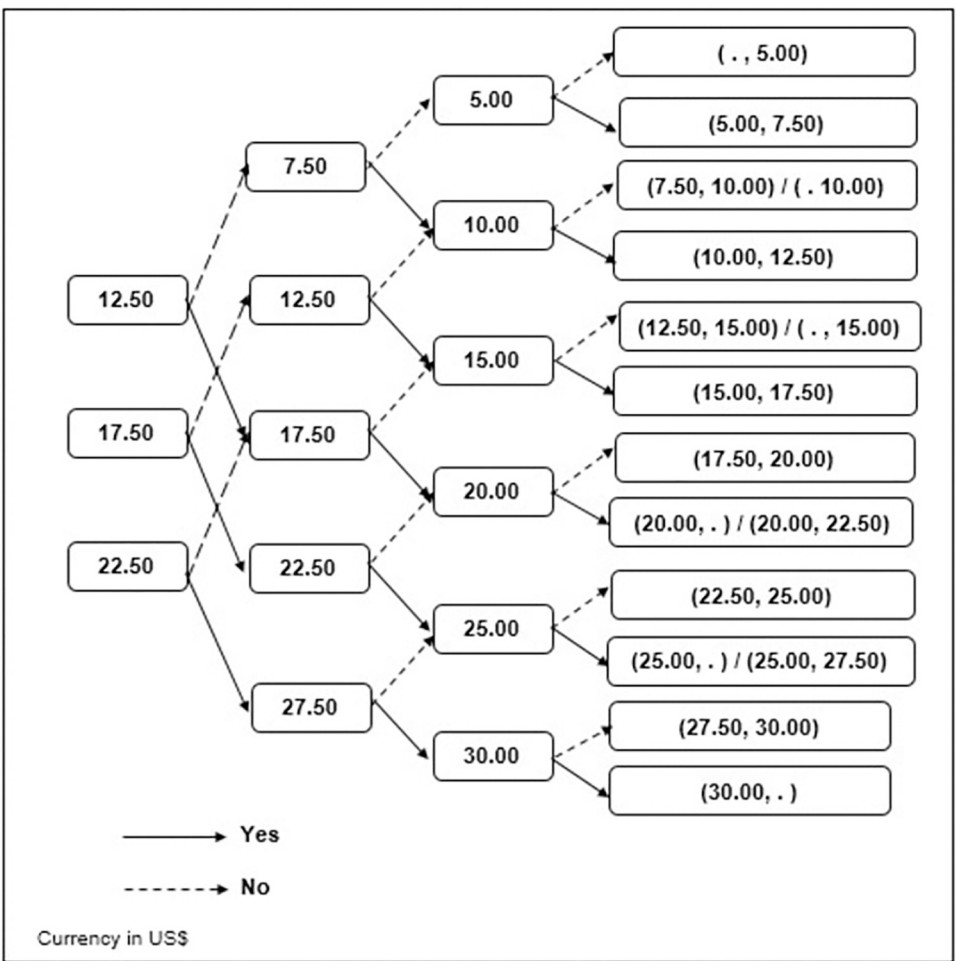

**Fig 1. Schematic representation of triple-bounded-dichotomous-choice experiment.** *Respondents were asked how much would they be willing to pay in additional taxes per year for any child who needs spectacles to have access to it.

the cost of a mixed offering to reflect the need for a combination of both spectacles types to address the refractive needs of the children.

## Data collection

Due to COVID-19 safety considerations, data were collected via telephone survey. To ensure data quality and consistency, the trained enumerator used a standard script when conducting the survey. Demographic details such as sex, age, and educational level were collected. Subsequently, three closed-ended questions were asked: (a) *"Are you willing to pay [an amount] in additional taxes per year for any child who needs spectacles to have access to it?"*, followed by (b) *"What if the amount is [an amount either higher/lower depending on previous response], would you be willing to pay?"*, followed by (c) *"And lastly, what if the amount is [an amount either higher/lower depending on previous response], would you be willing to pay?"*.

Fig 2 shows a questions route using the medium starting bid, US$17.50, as an example. We referred the approach adopted by Islam et al. in eliciting the final WTP [29]. If the respondent was willing to pay US$17.50 (responded Yes), a higher bid was offered at the second question–US$22.50; if the respondent was not willing to pay US$17.50 (responded No), a lower bid was

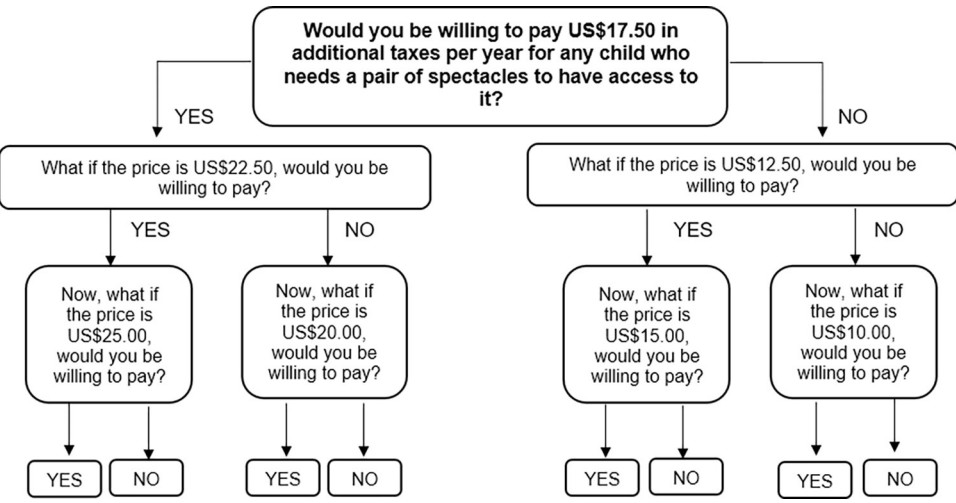

**Fig 2. Survey questions route, using starting bid of US$17.50 as an example.**

offered at the second question–US$12.50. The second and third question applied the same approach to arrive at the final WTP.

Considering that children required annual prescription changes, the survey questions were structured to ask taxpayers how much additional annual taxes they would be willing to offer to cover the reimbursement. The WTP estimation was assumed to be conservative because if the spectacles can last more than a year, the amount paid by the taxpayers would be exceeding the costs.

## Data management and statistical analysis

Statistical Package for the Social Sciences V25 (SPSS Inc., Chicago, IL) was used for data management and analysis. Data were cleaned and checked for consistency. Because we met the criterion of central limit theorem, parametric methods were adopted for its greater statistical power and ability to use 95% confidence intervals (CI) [31].

The study's primary outcome was the net benefits of a children's spectacle reimbursement scheme. The benefit in our study was determined using respondents' annual WTP for children's spectacles [32]. We assumed that the elements of benefit might include the aspect of additional lifetime income that can be attributed to the higher trajectory in earnings when the child is corrected with spectacles as URE has shown to have reduced the future income or increase children's educational inequalities [33]. The cost in our study was assumed to be that of ready-made spectacles, custom spectacles and a mix of these two types of spectacles in equal proportions. Net benefit was calculated by subtracting the cost of a pair of spectacles from the mean taxpayer's WTP. A positive net benefit means the benefit outweighs the cost, while a negative net benefit means that cost outweighs the benefit. Descriptive analysis was performed to obtain the mean WTP with standard deviation (SD) along with 95% CI. The differences in demographic characteristics among respondents in rural and urban Mongolia were tested using Chi-square test for categorical variables, and t-test for continuous variables, with a significance level of 5%.

The association of mean WTP with predictor variables, including geographic setting, age, sex, educational level, and children's RE status (those with a child or children prescribed spectacles due to RE versus those with children not needing spectacles), were assessed using t-test and ANOVA.

**Table 1. Demographic characteristics of survey respondents.**

| | Rural N (%) | Urban N (%) | Total N (%) | P-value comparing rural and urban respondents |
|---|---|---|---|---|
| Sex | | | | |
| Female | 22 (88.0%) | 17 (68.0%) | 39 (78.0%) | 0.172* |
| Male | 3 (12.0%) | 8 (32.0%) | 11 (22.0%) | |
| Age (years) | | | | |
| 20–35 | 12 (48.0%) | 11 (46.0%) | 23 (46.0%) | 0.862** |
| 36–50 | 10 (40.0%) | 9 (36.0%) | 19 (38.0%) | 0.411*** |
| $\geq$ 51 | 3 (12.0%) | 5 (20.0%) | 8 (16.0%) | |
| Mean Age ± SD (years) | 39.0 ± 10.2 | 41.3 ± 9.60 | 40.2 ± 9.86 | |
| Educational level | | | | |
| Illiterate | 1 (4.00%) | - | 1 (2.00%) | 1.00** |
| Primary | 2 (8.00%) | 1 (4.00%) | 3 (6.00%) | |
| Secondary | 11 (44.0%) | 12 (48.0%) | 23 (46.0%) | |
| Tertiary | 11 (44.0%) | 12 (48.0%) | 23 (46.0%) | |
| Total | 25 (100%) | 25 (100%) | 50 (100%) | |

* Yates Continuity Correction test was selected as a 2x2 table was assessed.

** Fisher's Exact Test was selected as expected cell values <5.

*** t-test.

## Results

### Participants' demographic profiles

All persons contacted (mean age 40.2 ± 9.86 years; 78.0% female) agreed to participate in the survey (n = 50, response rate = 100%). There was no statistical difference between respondents' mean age (p = 0.411), sex (p = 0.172), and educational levels (p = 1.00) in rural and urban settings. Among all respondents, more than two-thirds (84.0%) were below age 50 years, and over 90.0% completed either secondary or tertiary level education (Table 1).

### Cost-benefit analysis

The mean amount respondents were willing to pay in additional annual taxes for any child with RE to get a pair of free spectacles was US$24.00 ± 5.15 (95% CI US$22.55 to 25.35). Table 2 shows the calculation of net benefits. The calculations for ready-made spectacle revealed a positive net benefit of US$19.00. For custom spectacles, calculations found a negative net benefit of US$1.00. An analysis of the cost of mixed provision of spectacles found a positive net benefit of US$9.00, with benefits 1.6 times outweighing the cost.

### Factors associated with willingness-to-pay

Respondents of children with RE (US$22.50 ± 5.34) offered significantly less than those having children without RE (US$25.63 ± 4.50, p = 0.031). There was no significant difference between

**Table 2. Net benefits calculation.**

| | Ready-made spectacles | Custom spectacles | Mixed spectacles |
|---|---|---|---|
| Cost (US$) | 5.00 | 25.00 | 15.00 |
| Benefits* (US$) | 24.00 | 24.00 | 24.00 |
| Net Benefits (US$) | +19.00 | -1.00 | +9.00 |
| Benefits-to-Cost ratio | 4.8: 1.0 | 0.96: 1.0 | 1.6: 1.0 |

* As estimated by mean willingness-to-pay.

**Table 3. Potential predictors of mean willingness-to-pay (WTP).**

|  | Mean willingness-to-pay (WTP) US$ ± SD | P-value comparing groups |
|---|---|---|
| Setting | | |
| Rural | 24.30 ± 5.28 | 0.685* |
| Urban | 23.70 ± 5.11 | |
| Sex | | |
| Female | 23.46 ± 5.43 | 0.166* |
| Male | 25.91 ± 3.58 | |
| Age (years) | | |
| 20–35 | 24.67 ± 5.13 | 0.423** |
| 36–50 | 24.08 ± 4.58 | |
| ≥ 51 | 21.88 ± 6.51 | |
| Educational level | | |
| Illiterate | 15.00*** | 0.273** |
| Primary | 23.33 ± 7.64 | |
| Secondary | 23.59 ± 5.53 | |
| Tertiary | 24.89 ± 4.30 | |
| Children with refractive errors | | |
| No | 25.63 ± 4.50 | 0.031* |
| Yes | 22.50 ± 5.34 | |

* t-test.

** ANOVA.

*** Only one participant.

WTP of rural compared to urban respondents (p = 0.685), nor did WTP differ by age (p = 0.423), sex (p = 0.166) or educational level (p = 0.273) (Table 3).

## Discussion

We found a positive net benefit of US$9.00 in this CBA of a children's reimbursement scheme with equal probability of uptake of custom spectacles as opposed to ready-made spectacles. The mean WTP is independent of respondents' demographic characteristics, except children's RE status. Perhaps unexpectedly, parents of children without RE had significantly higher WTP in additional annual taxes for any child with RE to get spectacles than did parents of affected children might be influenced by variable such as income level that we did not include in the study.

The mean WTP in urban (US$23.70) and rural (US$24.30) settings in Mongolia are both higher than those found in a recent study assessing parental WTP for children's spectacles in Cambodia (US$18.60 and US$13.90 in the capital and rural settings, respectively) [34]. The observation of Cambodian respondents in offering lower WTP can be explained by the following reasons. Firstly, the proposed scheme in Mongolia was to include spectacle provision through the Social Health Insurance which will not incur any payment at the service point, while in Cambodia, the proposed cross-subsidisation scheme will require parents to pay a nominal amount. Secondly, the difference may be due to a higher gross domestic product per capita in Mongolia than Cambodia (US$4,339 versus US$1,643) [35,36]. Lastly, our study uses additional annual taxes as the payment vehicle, while in Cambodia, the payment was through out-of-pocket expenses.

Several studies demonstrate that exposure to "health shocks", such as the loss of vision associated with URE, can increase WTP [29,37]. In Cambodia, parents of children with refractive error were willing to pay a significantly higher amount (US$17.50 or more) than parents who

were unaware of their children's RE status [34]. Interestingly, in our study, respondents having children with RE had a lower WTP than respondents of children without. It may relate to unobserved heterogeneity related to income, for example, those with RE having lower income in our sample. This should be examined in further research with a larger sample and where details of income are collected.

Were only ready-made spectacles offered our study suggests a positive net benefit (US $19.00) while were only custom spectacles offered our study suggests a slightly negative net benefit (US$1.00). This suggests the benefits outweigh the costs of providing inexpensive ready-made spectacles but not custom spectacles. Concerning this, one proposed strategy would be for parents to "top-up" the government-subsidised spectacles when the cost of spectacles exceeds the subsidised amount. For example, if the government subsidises US$10.00 for any type of spectacles, parents will have to pay for the additional costs. This is especially referring to custom spectacles where the cost is often higher than the ready-made spectacles. A feasible structure of the reimbursement framework will be explored in our future research.

WTP has been widely used in the eye care sector to aid in service delivery planning, such as scheme for the cross-subsidisation of cataract surgery or spectacles [29,34,38]. We used WTP to estimate the potential benefits to inform a CBA, a novel approach in evaluating interventions related to children's URE. Due to limited resources and high demand for children's refractive services in Mongolia, policymakers must be informed of the value added by the intervention. This pilot CBA and future scale-up analysis should serve as a reference for the Mongolian government in making an informed policy decision.

The purpose of including children's spectacles into Social Health Insurance is to allow children who had URE access to spectacles without facing financial hardship, thus reducing the burden of VI due to URE. While we found no studies exploring the barriers to the provision of spectacles in Mongolia, based on the available literature, we assume that the following factors were associated with the significant burden. Firstly, Mongolia has a limited workforce that can deliver paediatric eye examination and spectacles dispensing [39]. Mountainous and upland steppe and semi-desert geography territories of Mongolia make children who live in rural unable to access eye health services and to procure spectacles in cities [40]. Lastly, approximately one-third of the Mongolian population was living below the poverty line [41], where the cost of spectacles might be a financial burden for them. To inform policymakers and advocate a reimbursement scheme for children's spectacles, we recommend exploration of these barriers should be included in the scale-up study.

Strengths of the current study include our having used a number of approaches recommended to increase the validity of the contingent valuation estimates: (i) using telephone interviews instead of surveys posted by mail; (ii) using WTP rather than willingness-to-accept; (iii) pretesting the survey before actual interviews; (iv) phrasing the WTP questions in a hypothetical scenario by indicating additional taxes that respondents would have to pay to subsidise free spectacles; and (v) collecting respondents' demographic characteristics [42].

Limitations of the study must also be acknowledged. The relatively small sample size may explain the lack of a significant association between WTP and most demographic factors. Secondly, hypothetical WTP surveys using contingent valuation may be prone to overestimation of the actual WTP amount [43]. Further, it has been suggested that a visual aid should be used when possible in WTP studies, so that biases due to miscomprehension can be avoided when participants are asked to make decisions about unfamiliar subjects [24]. We originally planned to present visual aids demonstrating to respondents the impact of spectacle wear. However, COVID-19 precautions made face-to-face interviews and the use of such aids impossible. In addition, employing the costs of ready-made spectacles and custom spectacles to construct the cost of a mixed offering may have also confounded the results, but we felt it was necessary to

reflect the local refractive needs. Finally, despite including respondents whose children participated in the vision screening programme in rural and urban schools (Orbis International's contact list), we did not include in our sample individuals not involved in the screening programme. This might cause selection bias and thus affecting the generalisability of the findings.

For recommendations, we suggest employing a random sampling method and sample size power calculation in future upscaling study. An open-ended final WTP question should also be included to obtain a more accurate estimation of the mean WTP and to address issues that might arise with censoring the maximum value at US$30. We only included variables such as parent's age, sex, educational level, resident location, and children's RE status in testing factors associated with WTP. The status of parent's income could be a key indicator that should be included, as demonstrated by other studies which found to be significantly correlated with the final WTP [34,44,45].

Despite its limitations, our analysis is one of the few examining the cost-benefit of national programmes providing spectacles for children. Our preliminary findings suggest that there is potential to include children's spectacles into the existing Social Health Insurance. However, further research with larger sample size is needed to confirm this.

## Supporting information

**S1 File. Database.**
(XLSX)

## Acknowledgments

We would like to thank Orbis International and Orbis Mongolia for the supports given.

## Author Contributions

**Conceptualization:** Ai Chee Yong, Chimgee Chuluunkhuu, Ving Fai Chan, Nathan Congdon, Ciaran O'Neill.

**Data curation:** Ai Chee Yong, Chimgee Chuluunkhuu, Ving Fai Chan.

**Formal analysis:** Ai Chee Yong, Ving Fai Chan, Ciaran O'Neill.

**Investigation:** Ai Chee Yong, Ving Fai Chan, Nathan Congdon, Ciaran O'Neill.

**Methodology:** Ai Chee Yong, Ving Fai Chan, Nathan Congdon, Ciaran O'Neill.

**Project administration:** Ai Chee Yong, Chimgee Chuluunkhuu.

**Supervision:** Ving Fai Chan, Nathan Congdon, Ciaran O'Neill.

**Validation:** Ai Chee Yong, Ciaran O'Neill.

**Visualization:** Ai Chee Yong, Chimgee Chuluunkhuu, Ving Fai Chan, Ciaran O'Neill.

**Writing – original draft:** Ai Chee Yong.

**Writing – review & editing:** Ai Chee Yong, Chimgee Chuluunkhuu, Ving Fai Chan, Tai Stephan, Nathan Congdon, Ciaran O'Neill.

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
