## [Decision Letter · Decision Letter 0]

29 Sep 2021

PONE-D-21-24356Cost-Benefit Analysis of a Children’s Spectacle Reimbursement Scheme: Evidence for Including Children’s Spectacles in Mongolia’s Social Health InsurancePLOS ONE

Dear Dr. Chan,

Thank you for submitting your manuscript to PLOS ONE. After careful consideration, we feel that it has merit but does not fully meet PLOS ONE’s publication criteria as it currently stands. Therefore, we invite you to submit a revised version of the manuscript that addresses the points raised during the review process.

This paper has been reviewed and the reviewers highlight a number of of strengths and weaknesses.

The authors should pay particular attention to Reviewer 1 comments, clarifying the study design and reflect on whether the commentary overreaches the actual scope of the work. Please also give further detail on the level of unmet visual need in Mongolia, and refractive services that are possible. This impacts on the question of the validity of ready-made spectacles as a means to meet demand in Mongolia. Finally, please given further information on determination of sample size.

We look forward to receiving your revised manuscript.

Kind regards,

Julie-Anne Little

Academic Editor

PLOS ONE

Journal Requirements:

2. Thank you for stating the following in the Competing Interests/Financial Disclosure * (delete as necessary) section: 

"I have read the journal's policy and the authors of this manuscript have the following competing interests: Professor Congdon declares that he is the Director of Research for Orbis International, Dr Chuluunkhuu declares that she is the Country Director for Orbis Mongolia, and Tai Stephan declares that she is the Global Programme Manager for Orbis International"

We note that you received funding from a commercial source:Orbis International

Please include your amended Competing Interests Statement within your cover letter. We will change the online submission form on your behalf

3. We note that you have referenced (ie. Bewick et al. [5]) which has currently not yet been accepted for publication. Please remove this from your References and amend this to state in the body of your manuscript: (ie “Bewick et al. [Unpublished]”) as detailed online in our guide for authors

Reviewers' comments:

Reviewer's Responses to Questions

**Comments to the Author**

1. Is the manuscript technically sound, and do the data support the conclusions?

Reviewer #1: Partly

Reviewer #2: Yes

2. Has the statistical analysis been performed appropriately and rigorously? 

Reviewer #1: Yes

Reviewer #2: Yes

3. Have the authors made all data underlying the findings in their manuscript fully available?

Reviewer #1: Yes

Reviewer #2: Yes

4. Is the manuscript presented in an intelligible fashion and written in standard English?

Reviewer #1: Yes

Reviewer #2: Yes

5. Review Comments to the Author

Reviewer #1: The cost-benefits analysis (CBA) of implementing the proposed cost reimbursement scheme for child’s spectacle sounds great. However, the paper should talk about the hypothetical WTP cost for spectacles and the saving it will make for the individual, society or government provided that there is a multi-tier spectacle scheme system in place, where capable pay higher charges for the same spectacle as they stated willing to pay while poor pay small amount. The detail of the scheme is important to discuss stating how this works for both families and government.

Having just WTP findings cannot hypothetically estimate the benefits of the proposed spectacle reimbursement scheme. How much money family/government save can be stated by WTP results, but it does not the give the monetary value of benefits (impacts of having the scheme)? CBA is not only about knowing the cost but also measuring the benefit by that cost. The later part is not covered in the paper. My question is why do authors need to say CBA? Why not just say a WTP study and report the findings and discuss the potential benefits?

If this is about the scheme to be launched by the government, it would be the compulsory service launched by the government in a public health system, which make richer pay high cost for the same product and poor pay less. If the parents would be willing to pay high amount of money for the spectacles, they would rather choose custom-made and high-quality spectacles that they can afford. Why would they need such scheme? The simple approach would be that the government can give the rebate if the annual income is low. Why to pay tax for the service or product which they may not require?

Yet one another question, if authors found parents are willing to pay high, why 90% of children do not have spectacles for URE in Mongolia (reference 17) as stated in the introduction. Literatures also suggest that the ready-made spectacles are useful and beneficial to small number of populations. Majority would need custom-made spectacles. The use and importance of ready-made spectacles are mainly for the resource poor settings where the refraction services and custom-made options are not available, inaccessible or unaffordable. Is this the situation in Mongolia?

To highlight the benefits and implication/ significance of the scheme, the paper should explain about the proposed child’s spectacles reimbursement scheme? Who is proposing this scheme and what are the details of this scheme should be explained in the introduction?

The sample size is too low to look at the association with the factors mentioned in the paper. I recommend referring this as a pilot study and not describing as a cost benefit analysis as the endpoint beneficiary group is not clear based on the inclusion of study participants (taxpayers only). As reported in strength of the current study in discussion - If additional taxes would have to be paid to subsidise free spectacles, why would these participants do so? They will/can purchase the spectacles straightforward rather than through the proposed scheme.

Design and setting section in methods state that WTP survey was used to estimate the benefits of providing spectacles to any children with RE. But the results only talk about cost of the product and WTP.

Sampling – target participants are taxpayers. Do you mean there are also people who do not pay tax because of low-income threshold? What is the income threshold level to whom this scheme is beneficial? If the problem is with those who cannot afford (who are not participants in the study), how this scheme would work is not clear?

How was sample size of 50 participants determined? It is very low for any epidemiological or population-based study. This number is ideal for a pilot study before a main or large sample study. Why not say this a pilot study (state in title and objective)? What do you mean to sampling distribution was assumed to be normal?

What is the basis for determining these bids figure $12.50, $17.50 or $22.50? It seems these were chosen to fit in the middle range of ready-made and custom-made spectacles. What proportions were ready to pay $12.50? What percentage were ready to pay $30? Why high- end of this bid is $30? It seems there are still high numbers of participants who would have been reporting that they are willing to pay high. Why not a last question was introduced to report the maximum amount they are willing to pay? It also sounds illogical to base on unpublished study.

Results/Discussion: The results are reported according to the methods mentioned. However, it should address the comments and questions raised above on the study methodology.

There is a positive net benefit of $9 to whom? Government or parents? How is this beneficial to those who are willing to pay higher price than average custom-made spectacles cost? Yes, the beneficiary group would be the lower income threshold families, who are not part of the study? It would have been clear if family income was collected, and income was analysed as a factor to relate the association.

Reviewer #2: Thank you for the opportunity to review this wonderful manuscript. This is well-constructed research with excellent writing. This study provides interesting new information to readers of PLOSONE and also to policymakers in Mongolia. I agree with authors on the strengths and acknowledge limitations of this study. Some specific comments noted below:

1. I am curious to see how the response will be if we explain the benefit of spectacles wear prior to questions about WTP since not everyone understands the importance of glasses.

2. Any other explanation on why parents of children have refractive error WTP is less than parents of children have no refractive error. Is this possible that those parents who have children with refractive error have better idea about the cost of glasses?

3. About the potential predictors of WTP, can parent’s income be one of the confounding factor?

4. "90% of Mongolian children with VI due to URE do not have spectacles". That number is significant, and the study focuses mainly on the cost-benefit of glasses. Is there anything else lead to this number, can it be accessibility of eye care services or something else? I think if we can address and recommend in future study that would be great resource for policymakers.

6. PLOS authors have the option to publish the peer review history of their article (what does this mean?). If published, this will include your full peer review and any attached files.

Reviewer #1: **Yes: **Prakash Paudel

Reviewer #2: **Yes: **Anh Vinh Bui

---

## [Author Response · Author response to Decision Letter 0]

9 Feb 2022

Responses to Academic Editor:

 1. The authors should pay particular attention to Reviewer 1 comments, clarifying the study design and reflect on whether the commentary overreaches the actual scope of the work. Response: We thank you for your suggestion. We addressed the above-mentioned concerns in our revised manuscript.

 2. Please also give further detail on the level of unmet visual need in Mongolia, and refractive services that are possible. This impacts on the question of the validity of ready-made spectacles as a means to meet demand in Mongolia.

Response: Published literature on the burden of unmet visual needs in Mongolia is limited. A study [1] showed that the prevalence of myopia among children in Mongolia was 5.8%, and according to the Resolution adopted by the Mongolian government on non-communicable diseases, 90% of children with refractive error did not own a pair of spectacles.[1] The burden of visual needs is exacerbated by the limited workforce for refractive services as there is limited optometry capacity in the country.[2] To address this challenge, ORBIS collaborated with the National Centre for Maternal and Child Health in Mongolia launched a 4-year programme (Model of Excellence in Modern Ophthalmology in Mongolia) to build local capacity by training ophthalmologists in urban and rural hospitals to perform refraction and dispense spectacles for children. We included the above information in the Introduction and Discussion.

3. Finally, please given further information on determination of sample size.

Response: The sample size was built upon the assumption that 30 or more participants will meet the principle of the central limit theorem, and therefore parametric test could be performed.[3] According to the central limit theorem, sampling distribution is considered normal with a minimum sample size of 30, in which sample mean will be closely gathered around the population mean. In Methods, we included a sentence like this - “According to the central limit theorem for sample size of more than 30, the sampling distribution was assumed to be normal. Upon discussions with the local researchers, we increase the sample size to 50 parents to avoid a high non-response rate.”

4. Thank you for stating the following in the Competing Interests/Financial Disclosure * (delete as necessary) section: 

"I have read the journal's policy and the authors of this manuscript have the following competing interests: Professor Congdon declares that he is the Director of Research for Orbis International, Dr Chuluunkhuu declares that she is the Country Director for Orbis Mongolia, and Tai Stephan declares that she is the Global Programme Manager for Orbis International". We note that you received funding from a commercial source: Orbis International. Please provide an amended Competing Interests Statement that explicitly states this commercial funder, along with any other relevant declarations relating to employment, consultancy, patents, products in development, marketed products, etc. 

Response: Thank you for highlighting this error. We have amended the submission.

5. We note that you have referenced (ie. Bewick et al. [5]) which has currently not yet been accepted for publication. Please remove this from your References and amend this to state in the body of your manuscript: (ie “Bewick et al. [Unpublished]”)

Response: Thank you for highlighting this. However, we did not cite Bewick et al.’s unpublished paper in our manuscript.

Responses to Reviewers:

Reviewer #1: 

Question 1:

The cost-benefits analysis (CBA) of implementing the proposed cost reimbursement scheme for child’s spectacle sounds great. However, the paper should talk about the hypothetical WTP cost for spectacles and the saving it will make for the individual, society or government provided that there is a multi-tier spectacle scheme system in place, where capable pay higher charges for the same spectacle as they stated willing to pay while poor pay small amount. The detail of the scheme is important to discuss stating how this works for both families and government.

Response: We thank the reviewer for the opportunity to clarify the perspective of the study as being that of the public payer and by extension the taxpayer.

Question 2:

Having just WTP findings cannot hypothetically estimate the benefits of the proposed spectacle reimbursement scheme. How much money family/government save can be stated by WTP results, but it does not the give the monetary value of benefits (impacts of having the scheme)? CBA is not only about knowing the cost but also measuring the benefit by that cost. The later part is not covered in the paper. My question is why do authors need to say CBA? Why not just say a WTP study and report the findings and discuss the potential benefits?

Response: We thank the reviewer for identifying a lack of clarity in our previous draft. As stated above the perspective of the study is that of the public payer and by extension the taxpayer over whose funds the payer has control. We accept that a range of benefits will flow from the scheme – benefits that extend beyond the improvements to visual acuity enjoyed directly by the individual whose refractive error is corrected and those which derive from this to include, for example, aspects of equity/solidarity. We have estimated the benefits to the taxpayer of the scheme using the study participant’s as representatives of the taxpayer. While we acknowledge this may give rise to risk of strategic bias – these are individuals whose children stand to benefit from the scheme - this must be counterbalanced by the risk of hypothetical bias associated with asking members of the public at large. We contend that the stated WTP provides an estimate of the present value of the discounted stream of benefits associated with the scheme expressed in monetary terms as perceived by taxpayers and as such can be compared with the schemes costs within a CBA. We have not sought to decompose or itemize the benefits but rather left this for the respondent to define in their own terms. We hope this clarifies this and have added additional text to the manuscript to provide further clarity and discuss the limitations of our approach. 

Question 3:

If this is about the scheme to be launched by the government, it would be the compulsory service launched by the government in a public health system, which make richer pay high cost for the same product and poor pay less. If the parents would be willing to pay high amount of money for the spectacles, they would rather choose custom-made and high-quality spectacles that they can afford. Why would they need such scheme? The simple approach would be that the government can give the rebate if the annual income is low. Why to pay tax for the service or product which they may not require?

Response: The reviewer identifies an interesting alternative to the scheme (i.e. cross-subsidisation) posited to participants and valued by them. As it is a distinct scheme, we recommend that the potential of such scheme to be researched further. 

Question 4:

Yet one another question, if authors found parents are willing to pay high, why 90% of children do not have spectacles for URE in Mongolia (reference 17) as stated in the introduction. 

Response: We thank the reviewer for his/her question which again points to the need for greater clarity. That parents are willing to pay for their own child is not what we sought to value. Rather we sought to value how much taxpayers would be willing to pay for any child in need to have access to spectacles. 

 We presumed factors such as ignorance of refractive error and limited access to optometry services could all contribute to the estimate that 90% of children do not have access to spectacles.

Question 5:

Literatures also suggest that the ready-made spectacles are useful and beneficial to small number of populations. Majority would need custom-made spectacles. The use and importance of ready-made spectacles are mainly for the resource poor settings where the refraction services and custom-made options are not available, inaccessible or unaffordable. Is this the situation in Mongolia?

Response: Thank you for highlighting this, and indeed Mongolia has minimal capacity in providing optometry services. We included ready-made spectacles in the analysis to highlight its effectiveness in resource-limited settings. According to the International Agency for the Prevention of Blindness (IAPB) country human resource database, there is no optometrist workforce in Mongolia.[2] The already low number of ophthalmologists in public hospitals were also deployed to deliver refractive services and spectacles dispensing for the children - another reason to the high burden of uncorrected refractive error in Mongolia. 

Question 6:

To highlight the benefits and implication/ significance of the scheme, the paper should explain about the proposed child’s spectacles reimbursement scheme? Who is proposing this scheme and what are the details of this scheme should be explained in the introduction?

Response: Thank you for highlighting this. As indicated above, our study focuses on cost-benefit analysis. The outcome of the findings can be used as an initial approach to propose to the government whether such a scheme is worth investing in reducing the burden of childhood vision impairment due to uncorrected refractive error in Mongolia. The designation of the reimbursement scheme will be our future work. We therefore added a description in the Introduction – “The Mongolian SHI is largely funded by the state central budget through general taxation. Should the proposed child’s spectacle reimbursement scheme be adopted by the government, the SHI would cover the reimbursement cost of eye examination and children’s spectacles. However, the actual framework for the reimbursement scheme is yet to be structured. The exploration of such a framework will be in our future scale-up study.”

Question 7:

The sample size is too low to look at the association with the factors mentioned in the paper. I recommend referring this as a pilot study and not describing as a cost benefit analysis as the endpoint beneficiary group is not clear based on the inclusion of study participants (taxpayers only). 

Response: We thank the reviewer for this suggestion. On reflection we are inclined to agree with the reviewer and have highlighted in our limitations section that the results of this small study should be treated with caution and that a larger study should be undertaken to investigate further our findings. 

Question 8:

As reported in strength of the current study in discussion - If additional taxes would have to be paid to subsidise free spectacles, why would these participants do so? They will/can purchase the spectacles straightforward rather than through the proposed scheme.

Response: We thank the reviewer for this question which we think arises from a misconception as to the role of the participant. Parents are being asked to value a scheme that benefits all children in need rather than just their own child. As noted, that is, the parent provides values as a taxpayer rather than a parent per se.

Question 9:

Design and setting section in methods state that WTP survey was used to estimate the benefits of providing spectacles to any children with RE. But the results only talk about cost of the product and WTP.

Response: We hope the comments provided already provide greater clarity on the approach adopted.

Question 10:

Sampling – target participants are taxpayers. Do you mean there are also people who do not pay tax because of low-income threshold? What is the income threshold level to whom this scheme is beneficial? If the problem is with those who cannot afford (who are not participants in the study), how this scheme would work is not clear?

Response: Hopefully our previous responses have clarified matters for the reviewer. To restate we sought to value the scheme from the public funder and by extension taxpayer’s perspective. As such a focus on the values of taxpayers seems entirely appropriate.

Question 11:

How was sample size of 50 participants determined? It is very low for any epidemiological or population-based study. This number is ideal for a pilot study before a main or large sample study. Why not say this a pilot study (state in title and objective)? What do you mean to sampling distribution was assumed to be normal?

Response: Thank you for raising this. We have addressed the question about sample size determination and sampling distribution in response to the academic editor’s comment above (kindly refer to Question 3). We acknowledge that 50 is a conservative number and therefore have amended in our title and objective that this is a pilot cost-benefit analysis. We edited the title, “A Pilot Cost-Benefit Analysis of a Children’s Spectacle Reimbursement Scheme: Evidence for Including Children’s Spectacles in Mongolia’s Social Health Insurance”; In the Introduction, “This pilot CBA is designed to assist eye health non-governmental organisation (NGO) Orbis International to provide initial findings to the Mongolian policymakers of the potential benefits of a proposed child’s spectacle reimbursement scheme.”

Question 12:

What is the basis for determining these bids figure $12.50, $17.50 or $22.50? It seems these were chosen to fit in the middle range of ready-made and custom-made spectacles. What proportions were ready to pay $12.50? What percentage were ready to pay $30? Why high- end of this bid is $30? It seems there are still high numbers of participants who would have been reporting that they are willing to pay high. Why not a last question was introduced to report the maximum amount they are willing to pay? It also sounds illogical to base on unpublished study.

Response: The bids were selected based on the cost of the spectacles and discussions with local eye care personnel as to their impressions of what might be reasonable. We agree a final open ended question would have been useful and would seek to incorporate that in further work.

The table below shows the proportion of final WTP by the surveyed respondents.

 Final Willingness-to-pay 

 US$12.50

n US$15.00

n US$17.50

n US$20.00

n US$22.50

N US$25.00

n US$27.50

n US$30.00

n Total n

Starting Bid US$12.5 1 4 1 2 8 0 0 0 16

 US$17.5 0 1 1 1 0 4 11 0 18

 US$22.5 0 1 0 1 1 2 1 10 16

Total (%) 1 (2.0%) 6 (12%) 2 (4.0%) 4 (8.0%) 9 (18%) 6 (12%) 12 (24%) 10 (20%) 50 (100%)

Question 13:

Results/Discussion: The results are reported according to the methods mentioned. However, it should address the comments and questions raised above on the study methodology.

Response: Please see above our responses to the feedback.

Question 14:

There is a positive net benefit of $9 to whom? Government or parents? How is this beneficial to those who are willing to pay higher price than average custom-made spectacles cost? Yes, the beneficiary group would be the lower income threshold families, who are not part of the study? It would have been clear if family income was collected, and income was analysed as a factor to relate the association.

Response: From a public payer’s perspective – hopefully this is now clearer.

 We agree that family income would be useful to help ascertain the face validity of the values generated, however, we did not include this as one of the demographic variables. We have therefore included this as a weakness in the Discussion.

Reviewer #2: 

Thank you for the opportunity to review this wonderful manuscript. This is well-constructed research with excellent writing. This study provides interesting new information to readers of PLOSONE and also to policymakers in Mongolia. I agree with authors on the strengths and acknowledge limitations of this study. Some specific comments noted below:

Question 1:

I am curious to see how the response will be if we explain the benefit of spectacles wear prior to questions about WTP since not everyone understands the importance of glasses.

Response: Thank you for the comment. We agreed that not all participants understand the benefit of spectacle wear, especially those without an uncorrected refractive error. However, we did not brief the participants about the benefit of spectacles wear prior to the WTP questions because rather than seek to itemise and describe the benefits which could intentionally influence the respondent, we chose to allow them to base responses on their own perceptions of benefits.

Question 2:

Any other explanation on why parents of children have refractive error WTP is less than parents of children have no refractive error. Is this possible that those parents who have children with refractive error have better idea about the cost of glasses?

Response: Thank you for the question. We explained in the Discussion - “Interestingly, in our study, respondents having children with refractive error had a lower WTP than respondents of children without. This may be because these parents’ have a different appreciation as to the benefit derived by children from the use of spectacles but we can only speculate as to what may underlie this result. It may relate to unobserved heterogeneity related to income, for example, those with RE having lower income in our sample. This should be examined in further research with a larger sample and where details of income are collected.” 

Question 3:

About the potential predictors of WTP, can parent’s income be one of the confounding factor?

Response: Thank you for highlighting this. We agree that parent’s income is potentially a confounding factor that we were unable to address. We have therefore acknowledged this as one of the limitations in the Discussion and recommended to include this in the future study - “We only included variables such as parent’s age, sex, educational level, resident location, and children’s RE status in testing factors associated with WTP. The status of parent’s income could be a key indicator that should be included, as demonstrated by other studies which found to be significantly correlated with the final WTP.”

Question 4:

"90% of Mongolian children with VI due to URE do not have spectacles". That number is significant, and the study focuses mainly on the cost-benefit of glasses. Is there anything else lead to this number, can it be accessibility of eye care services or something else? I think if we can address and recommend in future study that would be great resource for policymakers.

Response: Thank you for the suggestion. We added a sentence in the Discussion – “The purpose of including children’s spectacles into Social Health Insurance is to allow children who had URE can access spectacles without facing financial hardship, thus reducing the burden of VI due to URE. While we found no studies exploring the barriers to the provision of spectacles in Mongolia, based on the available literature, we assume that the following factors were associated with the significant burden. Firstly, Mongolia has a limited workforce that can deliver paediatric eye examination and spectacles dispensing. Mountainous and upland steppe and semi-desert geography territories of Mongolia make children who live in rural unable to access eye health services and to procure spectacles in cities. Lastly, approximately one-third of the Mongolian population was living below the poverty line, where the cost of spectacles might be a financial burden for them. To inform policymakers and advocate a reimbursement scheme for children’s spectacles, we recommend exploration of those barriers should be included in the scale-up study.”

References

1. Mongolia Government. Mongolia National Program for Non-Communicable Diseases - The Government Resolution No.34 “Adoption of the National Programme". 289 Mongolia; 2017. 

2. Number of Eye Care Personnel in Mongolia. [cited 5 Nov 2021]. Available: https://www.iapb.org/learn/vision-atlas/magnitude-and-projections/countries/mongolia

3. Kwak SG, Kim JH. Central limit theorem: the cornerstone of modern statistics. Korean J Anesth. 2017;70: 144–156.

---

## [Decision Letter · Decision Letter 1]

12 Apr 2022

PONE-D-21-24356R1A Pilot Cost-Benefit Analysis of a Children’s Spectacle Reimbursement Scheme: Evidence for Including Children’s Spectacles in Mongolia’s Social Health InsurancePLOS ONE

Dear Dr. Chan,

Thank you for submitting your manuscript to PLOS ONE. After careful consideration, we feel that it has merit but does not fully meet PLOS ONE’s publication criteria as it currently stands. Therefore, we invite you to submit a revised version of the manuscript that addresses the points raised during the review process. Please review and respond to the comments of reviewer 1 below.

We look forward to receiving your revised manuscript.

Kind regards,

Julie-Anne Little

Academic Editor

PLOS ONE

Additional Editor Comments:

Thanks to the authors for addressing the majority of the reviewers comments. Please review and address the remaining issues from one of the reviewers.

Reviewers' comments:

Reviewer's Responses to Questions

**Comments to the Author**

1. If the authors have adequately addressed your comments raised in a previous round of review and you feel that this manuscript is now acceptable for publication, you may indicate that here to bypass the “Comments to the Author” section, enter your conflict of interest statement in the “Confidential to Editor” section, and submit your "Accept" recommendation.

Reviewer #1: All comments have been addressed

Reviewer #2: All comments have been addressed

2. Is the manuscript technically sound, and do the data support the conclusions?

Reviewer #1: Partly

Reviewer #2: Yes

3. Has the statistical analysis been performed appropriately and rigorously? 

Reviewer #1: Yes

Reviewer #2: I Don't Know

4. Have the authors made all data underlying the findings in their manuscript fully available?

Reviewer #1: Yes

Reviewer #2: Yes

5. Is the manuscript presented in an intelligible fashion and written in standard English?

Reviewer #1: Yes

Reviewer #2: Yes

6. Review Comments to the Author

Reviewer #1: It is good to read the revised manuscript with acknowledgement of the limitations on the generalisability of the results due to small sample size and no income threshold data to relate the validity of the findings. As indicated (with $19 benefit for RMS), the results suggest relatively high WTP of Mongolians despite 90% RE children (relating half suitable for ready-made) being possibly uncorrected. In addition to this, those with RE are more likely to pay less than those without, is certainly question of investigation. As concluded, the full-scale study with greater sample size and proper selection of participants, will hopefully contribute to purposing the structure of reimbursement for spectacle.

Unlike this study (spectacle reimbursement scheme), previous WTP studies contribute/suggest for “spectacle cost-subsidisation scheme”. You may discuss the difference and highlight the SMI features and focus. Importantly, is this meant for RMS, custom-made or both? The question and response are dependent to the specific product and accordingly relate the scheme. Ideally, WTP question is to base on the focus or the need; for ready-made or custom-made. If relevant and useful, add to highlight.

Discussion third paragraph - Unlike this study, reference no 34 do not have a compare group. The Cambodia study included only parents of children with RE. Please correct ‘an eye disorder’ with ‘refractive error’. About 53% parents were willing to pay $17.50 (standard price of custom-made spectacle) or more.

Reason for “Those with RE had low WTP compared to without” is simply unanswerable in given situation. The listed potential reasons (different appreciation and income) do not quite relate or suit. I suggest delete and simply indicate the need for clarification /verification through full-scale study.

In the concluding statements, it is better to state clearly about the scheme that it is the proposed or potential spectacle reimbursement scheme under SMI.

Reviewer #2: (No Response)

7. PLOS authors have the option to publish the peer review history of their article (what does this mean?). If published, this will include your full peer review and any attached files.

Reviewer #1: **Yes: **Prakash Paudel

Reviewer #2: No

---

## [Author Response · Author response to Decision Letter 1]

3 May 2022

Reviewer #1: 

It is good to read the revised manuscript with acknowledgement of the limitations on the generalisability of the results due to small sample size and no income threshold data to relate the validity of the findings. As indicated (with $19 benefit for RMS), the results suggest relatively high WTP of Mongolians despite 90% RE children (relating half suitable for ready-made) being possibly uncorrected. In addition to this, those with RE are more likely to pay less than those without, is certainly question of investigation. As concluded, the full-scale study with greater sample size and proper selection of participants, will hopefully contribute to purposing the structure of reimbursement for spectacle.

1. Unlike this study (spectacle reimbursement scheme), previous WTP studies contribute/suggest for “spectacle cost-subsidisation scheme”. You may discuss the difference and highlight the SMI features and focus. Importantly, is this meant for RMS, custom-made or both? The question and response are dependent to the specific product and accordingly relate the scheme. Ideally, WTP question is to base on the focus or the need; for ready-made or custom-made. If relevant and useful, add to highlight.

Response: Thank you for suggesting this. We added this into our Discussions: “The observation of Cambodian respondents in offering lower WTP can be explained by the following reasons. Firstly, the proposed scheme in Mongolia was to include spectacle provision through the Social Health Insurance which will not incur any payment at the service points, while in Cambodia, the proposed cross-subsidisation scheme will require parents to pay a nominal amount. Secondly, the difference may be due to a higher gross domestic product per capita in Mongolia than Cambodia (US$4,339 versus US$1,643).[35,36] Lastly, our study uses additional annual taxes as the payment vehicle, while in Cambodia, the payment was through out-of-pocket expenses.”

 We recognised the limitation of not specifying which types of spectacles (ready-made or custom-made) that the tax payer would be willing to pay. However, we chose to use mix spectacles (US$15) as the spectacle cost because i) one in two Mongolian children who had refractive error can be corrected effectively from a pair of inexpensive, ready-made spectacles, and ii) to reflect the practical situation where there will be need to have a combination of both ready-made and custom-made spectacles to cover the refractive needs of the children. “Based on a dataset obtained from an outreach vision screening programme initiated by an international eye NGO, one in two Mongolian children who had refractive error can be corrected effectively from a pair of inexpensive, ready-made spectacles. Therefore, the costs of ready-made spectacles and custom spectacles were used to construct the cost of a mixed offering to reflect the need for a combination of both spectacle types to address the refractive needs of the children.” We included this content in the Methods section.

 We also highlighted the following in the Limitations “In addition, employing the costs of ready-made spectacles and custom spectacles to construct the cost of a mixed offering may have also confounded the results, but we felt it was necessary to reflect the local refractive needs.”

2. Discussion third paragraph - Unlike this study, reference no 34 do not have a compare group. The Cambodia study included only parents of children with RE. Please correct ‘an eye disorder’ with ‘refractive error’. About 53% parents were willing to pay $17.50 (standard price of custom-made spectacle) or more.

Response: Thank you for spotting this error. We have amended the sentence accordingly.

3. Reason for “Those with RE had low WTP compared to without” is simply unanswerable in given situation. The listed potential reasons (different appreciation and income) do not quite relate or suit. I suggest delete and simply indicate the need for clarification /verification through full-scale study.

Response: Thank you for highlighting this. We agreed that findings of “those with RE had low WTP compared to without” is unanswerable. We also agreed that there is a need for clarification/verification through full-scale study, as added to the Discussion “This should be examined in further research with a larger sample and where details of income are collected”.

4. In the concluding statements, it is better to state clearly about the scheme that it is the proposed or potential spectacle reimbursement scheme under SMI.

Response: Thank you for highlighting this. In the conclusion, we made clear statement that, “Our preliminary findings suggest that there is potential to include children’s spectacles into the existing Social Health Insurance. However, further research with larger sample size is needed to confirm this.”

Reviewer #2: (No Response)

---

## [Decision Letter · Decision Letter 2]

2 Aug 2022

A Pilot Cost-Benefit Analysis of a Children’s Spectacle Reimbursement Scheme: Evidence for Including Children’s Spectacles in Mongolia’s Social Health Insurance

PONE-D-21-24356R2

Dear Dr. Chan,

We’re pleased to inform you that your manuscript has been judged scientifically suitable for publication and will be formally accepted for publication once it meets all outstanding technical requirements.

Kind regards,

Julie-Anne Little

Academic Editor

PLOS ONE

Additional Editor Comments (optional):

Both reviewers were satisfied that you have addressed their comments.

Reviewers' comments:

Reviewer's Responses to Questions

**Comments to the Author**

1. If the authors have adequately addressed your comments raised in a previous round of review and you feel that this manuscript is now acceptable for publication, you may indicate that here to bypass the “Comments to the Author” section, enter your conflict of interest statement in the “Confidential to Editor” section, and submit your "Accept" recommendation.

Reviewer #1: All comments have been addressed

Reviewer #2: (No Response)

2. Is the manuscript technically sound, and do the data support the conclusions?

Reviewer #1: Yes

Reviewer #2: Yes

3. Has the statistical analysis been performed appropriately and rigorously? 

Reviewer #1: Yes

Reviewer #2: Yes

4. Have the authors made all data underlying the findings in their manuscript fully available?

Reviewer #1: Yes

Reviewer #2: Yes

5. Is the manuscript presented in an intelligible fashion and written in standard English?

Reviewer #1: Yes

Reviewer #2: Yes

6. Review Comments to the Author

Reviewer #1: (No Response)

Reviewer #2: It will be beneficial for the next study to use Visual Analog Scale to evaluate WTP. Looks like from the WTP of both rural and urban people is close to custom lenses price so maybe other factor like accessibility to eye care is bigger issue. Also the cost of ready made and custom made glasses might be lowered when the this scheme is adapted to the government coverage since they will have better buying power to negotiate better price.

7. PLOS authors have the option to publish the peer review history of their article (what does this mean?). If published, this will include your full peer review and any attached files.

Reviewer #1: **Yes: **Prakash Paudel

Reviewer #2: No

---

## [Editor Report · Acceptance letter]

5 Aug 2022

PONE-D-21-24356R2 

A Pilot Cost-Benefit Analysis of a Children’s Spectacle Reimbursement Scheme: Evidence for Including Children’s Spectacles in Mongolia’s Social Health Insurance 

Dear Dr. Chan:

I'm pleased to inform you that your manuscript has been deemed suitable for publication in PLOS ONE. Congratulations! Your manuscript is now with our production department. 

Kind regards, 

on behalf of

Dr. Julie-Anne Little 

Academic Editor

PLOS ONE